# Recurrent Uveitis-Glaucoma-Hyphema Syndrome Due to Positional Pupillary Capture after Sutureless Scleral-Fixated Secondary Intraocular Lens Placement

Sagar Patel [1] and Hossein Ameri [2,*]

1    Retina Consultants of Texas, Houston, TX 77339, USA
2    Department of Ophthalmology, USC Roski Eye Institute, Keck School of Medicine, University of Southern California, Los Angeles, CA 90033, USA
*    Correspondence: ameri@med.usc.edu

**Abstract:** Here, we present a case of floppy iris leading to positional pupillary capture by a sutureless, scleral-fixated intraocular lens (IOL) causing recurrent uveitis-glaucoma-hyphema (UGH) syndrome. The patient developed recurrent episodes of UGH syndrome after dislocated IOL removal and the placement of sutureless, scleral-fixated IOL. Gravitationally dependent pupillary capture was noted with the superior iris moving in front of and behind the IOL, depending on head positioning. Ultrasonography showed a floppy iris that moved with shifting gaze. The lack of the capsular bag may have contributed to extreme iris movements. This finding may be secondary to a combination of a lack of zonular support and capsular bag support as well as the lack of vitreous support following vitrectomy. When possible, secondary IOL placement behind a peripherally preserved capsular bag may reduce the risk of UGH.

**Keywords:** scleral-fixated intraocular lens; uveitis-glaucoma-hyphema syndrome; pupillary capture; secondary intraocular lens; iris capture

## 1. Introduction

Several approaches for secondary intraocular lens (IOL) placement have been described. The ideal approach to IOL placement is in the bag or sulcus, as these have the best visual outcomes and the fewest complications [1]. Without adequate capsular support, options include an anterior chamber IOL, iris-fixated IOL, or scleral-fixated IOL. Each procedure has advantages and disadvantages [2]. The technique chosen depends on the ocular morbidities and surgeon's preference.

No clear consensus exists on which technique is best [3]. One concern with scleral sutured IOLs is suture breakage or extrusion [4]. Due to this concern, the approach was modified with sutureless scleral-fixation of a PCIOL. Both techniques have similar visual outcomes and complication rates [5]. However, given the novelty of the sutureless technique, specific complications are not known.

We report a case of floppy iris leading to positional pupillary capture by a sutureless, scleral-fixated IOL causing recurrent uveitis-glaucoma-hyphema (UGH) syndrome and vitreous hemorrhage.

## 2. Case Report

A 79 year-old female presented with blurred vision in the right eye was noted to have a nasally displaced IOL leading to visually significant diplopia. She had a Staar AA-4203 lens placed in 1994. She did not have any history of glaucoma or pseudoexfoliation syndrome pre-operatively. She underwent pars plana vitrectomy and posterior chamber IOL exchange with sutureless scleral fixation. A complete vitrectomy was performed and the remaining capsular bag was removed. The IOL was removed through corneal incision. A 20.5 diopter

MA60AC IOL was fixated to the sclera 3 mm posterior to the limbus using the modified Yamane technique [6].

On post-operative month one, the superotemporal iris was noted to be behind the IOL (Figure 1). Examination also showed significant iris movement when the patient looked up and down without IOL movement. Interestingly, when the patient was asked to bend forward, the iris moved to the front of the IOL but when the exam chair reclined backwards, the iris moved to behind the IOL. Ultrasound biomicroscopy showed iridodonesis and IOL-chafing at 10:30 (Figure 2). The superior iris was noted to have significant movement when the eye moved from superior gaze to inferior gaze. The IOL was slightly decentered and tilted at 10:30, but was stable.

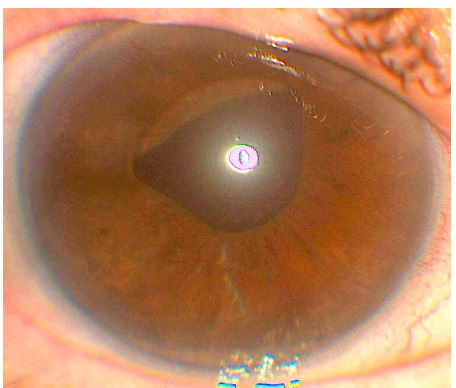

**Figure 1.** Slit lamp photo demonstrating pupillary capture by the IOL superotemporaly.

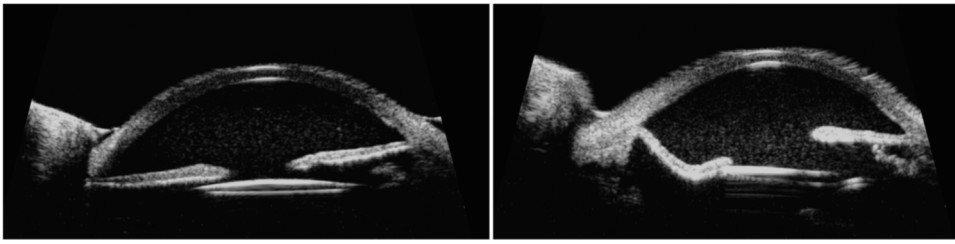

**Figure 2.** Anterior segment ultrasound biomicroscopy showing the iris and IOL in primary position (**left**) and marked iridodonesis, posterior-iris-bowing, and IOL-chafing when the eye moved down (**right**).

The floppy iris resulted in multiple episodes of elevated IOP. The IOP ranged from 9 mmHg on the lower end between episodes and to a maximum IOP of 46 mmHg during the episodes. The IOP spikes were treated with dorzolamide, timolol, brimonidine, and latanoprost eye drops. Her visual acuity ranged from 20/25 between the episodes to 20/1000 to hand motions during the episodes. Pilocarpine 1% was administered to keep the iris in front of the IOL and prevent recurrent chafing, but the patient continued to have episodes of uveitis-glaucoma-hyphema syndrome and pupillary capture.

Iridoplasty versus removal of the IOL were discussed with the patient and she opted for the latter. The IOL was successfully explanted and the patient left aphakic. Following the IOL removal, her IOP remained within normal limits without the use of medications and her visual acuity was 20/25 with an aphakic contact lens.

## 3. Discussion

Secondary IOL placement is a useful technique to address complications of cataract surgery, including dislocated IOLs. Complications associated with scleral-fixated IOLs include hemorrhage (hyphema or vitreous), cystoid macular edema, glaucoma, corneal edema, uveitis, and retinal detachment [1]. UGH syndrome has been reported to occur in 3% of cases after sutured scleral-fixated IOLs [7]. Intermittent pupillary capture of the IOL

optic has been reported in 7.9% of these patients [8]. Most studies analyzing outcomes and complications of scleral-fixated IOLs have studied scleral sutured IOLs. These studies have shown that long-term complications related to the suture remain a problem [9]. Newer techniques have been developed to reduce suture-related complications by using sutureless scleral fixation [6]. Given the recent advent of this technique, unforeseen complications may exist.

We present a patient with recurrent episodes of UGH syndrome due to floppy iris leading to positional pupillary capture after sutureless scleral fixation of the IOL. We noted that the iris moved posterior to the IOL optic while the patient was supine, and the iris moved anterior to the IOL when the patient bent forward. The ultrasound biomicroscopy showed the iris to be significantly floppy with a wide range of movement when shifting between up and down gaze, repeatedly chafing on the IOL. While the IOL showed a slight tilt, it did not show any movement. This gravitationally dependent pupillary capture may have resulted in episodes of iris chafing that worsened when the patient was supine, leading to recurrent UGH syndrome. When the pupil was constricted using pilocarpine, the episodes continued. Another possible way to explain this could be that the pupillary capture was due to the passage of the haptics through a portion of the posterior iris tissue.

While pupillary capture has been described in up to 7.9% of patients after scleral sutured IOL, we present the first case of gravitationally dependent pupillary capture from floppy iris after sutureless secondary IOL placement. We suspect that the lack of zonules and capsular bag support as well as the lack of vitreous support following the vitrectomy may have contributed to the extreme iris movements in this case, leading to the pupillary capture. Another consideration is the placement of the haptics 3 mm from the limbus. A more anterior placement of 2 mm may have supported the iris better and limited its floppiness. Previous reports have found floppy iris to occur following scleral sutured IOL implantation in patients with Marfan syndrome [10]. Our case is unique due to the gravitational dependence of the pupillary capture. It is plausible that intraocular fluid flow around the IOL during the movement of the eye between an upright and supine position may result in the extreme movement of the iris due to the lack of capsular bag and zonular support posterior to the iris (Figure 3). If the capsular bag and zonules were intact, the vitreous fluid motion would not be transmitted to the iris. We suspect that when the eye was moved, the angular momentum of the fluid within the eye exerted a force upon the iris causing the positional pupillary capture. When examining the ultrasound biomicroscopy video, the iris exhibited a sporadic and sinusoidal floppiness with the movement of the eye.

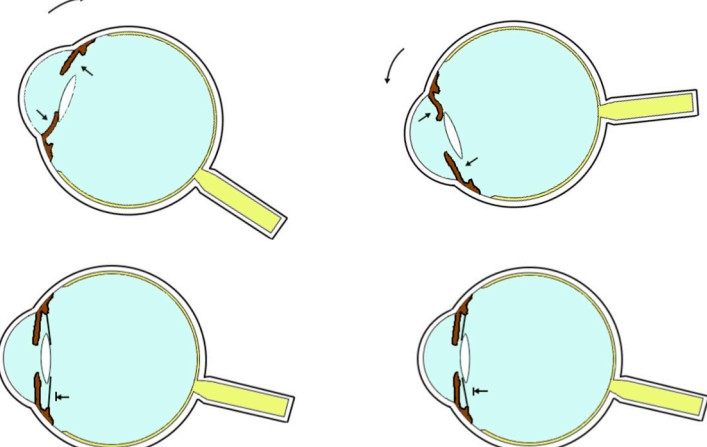

**Figure 3.** Model demonstrating aqueous flow with eye movement leading to anterior movement of the superior iris with up gaze (**top left**), posterior movement of the superior iris with down gaze (**top right**), and blockage of aqueous flow when the capsular bag is in place (shown in black) behind the IOL (**bottom left**) or in front of the IOL (**bottom right**). Note the haptics of the IOL are not shown.

The sutureless technique remains a useful and effective approach due to a lack of suture-related complications. As with any surgery, the risks and benefits of sutureless surgery must be considered. There are multiple other IOL fixation options that also must be considered including sutured IOL, iris fixated IOL, anterior chamber IOL, and aphakic contact lens. The lack of capsular bag support may have contributed to the extreme iris movement in this case. When possible, preservation of zonules and peripheral capsular bag and placement of the secondary IOL in front of or behind the partially preserved capsular bag may prevent this complication. If the IOL is going to be placed behind the capsular bag, a large central capsulotomy is needed to allow the passage of the IOL. Prior to placing a sutureless, scleral-fixated IOL, the surgeon should examine the patient for any signs of floppy iris. Proper patient selection is key to successful secondary sutureless IOL placement.

**Author Contributions:** Conceptualization, H.A.; methodology, H.A., S.P.; validation, H.A. and S.P.; formal analysis, H.A. and S.P.; investigation, H.A. and S.P.; data curation, H.A. and S.P. ; writing—S.P.; writing—review and editing, H.A.; visualization, S.P.; supervision, H.A.; project administration, H.A. and S.P. All authors have read and agreed to the published version of manuscript.

**Funding:** This research was funded by an Unrestricted Grant to the Department of Ophthalmology from Research to Prevent Blindness, New York, NY.

**Institutional Review Board Statement:** Ethical review and approval were waived for this study due to REASON the retrospective nature of the project.

**Informed Consent Statement:** Patient consent was waived due to the retrospective nature of the project.

**Data Availability Statement:** Not applicable.

**Conflicts of Interest:** The authors declare no conflict of interest.

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
