# Peer review of "Recurrent Uveitis-Glaucoma-Hyphema Syndrome Due to Positional Pupillary Capture after Sutureless Scleral-Fixated Secondary Intraocular Lens Placement"

_2813-1053, doi:10.3390/jcto1010003_

Round 1

Reviewer 1 Report

The case presented by Patel and Ameri is a unique complication of the sutureless scleral fixed IOL technique.

Could it be that there was some connection made between the posterior iris and the haptic?  Could the complication have been the haptic passing through some portion of the posterior iris tissue? The SL photos does not illustrate the haptic insertion sites and has been cropped. I believe that with the floppy iris, this alternative hypothesis makes more sense than a simple gravitational model proposed.

Overall the level of imaging detail is insufficient. Where are the foot plates of the haptics in the sclera? Can these be illustrated? Can more cuts of the OCT be provided to exclude the possibility that the haptic has somehow engaged the posterior iris surface enabling this very odd engagement of the tissue? I do not believe that a floppy iris would obtain or maintain such a configuration without being fixed to the IOL. If so the mechanism of the complication in this case is a very different than the one presented.

Many patients do not have a capsule and do not have extremely floppy iris. Though I agree with some of the hypotheses of the authors, perhaps their hypothesis should be restated or qualified as one of several possibilities? Furthermore, the statement that placement of such a lens behind a capsular bag may help reduce such complications is written in such a way to suggest that surgeons have a choice of doing so when in reality most patients who present with the need for scleral fixed lenses will have already lost their capsules at the time of secondary IOL implantation! Again, though I generally agree with the point made by the authors, I would rephrase the presentation. I would list all three factors cited in discussion in the abstract, ie. a combination of lack of capsule, associated zonules, and hyaloid face loss after a vitrectomy

The point about fluid flow in a unicameral post-vitrectomy eye is not correct. Only the tissue are differentially subjected to gravity, I very much doubt the impact of gravity changes on the fluids in the eye!  Please explain further

I am not certain that moving the position of the lens anteriorly by 1 mm (2mm posterior) would have necessarily made it less likely to engage iris.

In the results section of the case description the multiple episodes of OHT is a bit unclear. Understanding the IOP range between episodes as well as the OHT range during capture should be spelled out more clearly. The lower end of the range ("9") is NOT an example of ocular hypertension. Units mmHg.

"As with any surgery, the risks and benefits of sutureless surgery must be considered" -- what are the choices for patients -- suture, suture-less, ACIOL and aphakic CL. List these more clearly.

Minor comments:

First sentence is not a complete sentence in the abstract (!)

I agree with the authors that exam for floppy iris in important before Yamane undertaken, but how exactly is this done? What are the alt choice for surgery that should be advocated if a floppy iris is identified? Is this complication common enough to justify a different IOL placement technique and if so at what threshold? 

Figure 3 needs revision as gazes are too similar to illustrate the point. If quantifiable (degrees), all the better to add

Author Response

Could it be that there was some connection made between the posterior iris and the haptic?  Could the complication have been the haptic passing through some portion of the posterior iris tissue? The SL photos does not illustrate the haptic insertion sites and has been cropped. I believe that with the floppy iris, this alternative hypothesis makes more sense than a simple gravitational model proposed.

Response: The haptics did not pass through any portion of the posterior iris tissue. As mentioned, the haptics were fixated to the scleral 3 mm posterior to the limbus. Unfortunately, we do not have any other split lamp photos.

Overall the level of imaging detail is insufficient. Where are the foot plates of the haptics in the sclera? Can these be illustrated? Can more cuts of the OCT be provided to exclude the possibility that the haptic has somehow engaged the posterior iris surface enabling this very odd engagement of the tissue? I do not believe that a floppy iris would obtain or maintain such a configuration without being fixed to the IOL. If so the mechanism of the complication in this case is a very different than the one presented.

Response: We do not have other photos to demonstrate the extruded haptic ends. We do not have any further OCT cuts through the iris but we are certain that iris tissue was not engaged. As mentioned in the case report - he superior iris was noted to have significant movement when the eye moved from superior gaze to inferior gaze. The IOL was slightly decentered and tilted at 10:30, but was stable. Given that the IOL was stable while the iris moved, it is unlikely that the iris tissue was engaged by the IOL haptics. If this was the case, then the IOL should have been unstable as well. Also Figure 2 shows the clear bowing of the iris upon downgaze with anterior displacement of the opposite end of the iris. This configuration would not be caused by haptic engagement of the iris surface and suggests that a force is acting upon the iris in opposite directions.

Many patients do not have a capsule and do not have extremely floppy iris. Though I agree with some of the hypotheses of the authors, perhaps their hypothesis should be restated or qualified as one of several possibilities? Furthermore, the statement that placement of such a lens behind a capsular bag may help reduce such complications is written in such a way to suggest that surgeons have a choice of doing so when in reality most patients who present with the need for scleral fixed lenses will have already lost their capsules at the time of secondary IOL implantation! Again, though I generally agree with the point made by the authors, I would rephrase the presentation. I would list all three factors cited in discussion in the abstract, ie. a combination of lack of capsule, associated zonules, and hyaloid face loss after a vitrectomy

 Response: We have clarified in the discussion that we present one possible hypothesis and that there may be other ways to explain this. We have clarified in the paper that preservation of the capsular bag when possible may help. These three factors cited in discussion have been added to the abstract.

The point about fluid flow in a unicameral post-vitrectomy eye is not correct. Only the tissue are differentially subjected to gravity, I very much doubt the impact of gravity changes on the fluids in the eye!  Please explain further.

Reponse: We are not saying that the fluid flow is due to gravity. When the eye is rotated up and down, the angular momentum of the fluid within the eye does exert some force upon the iris. This may be a contributing factor to the pupillary capture. Our hypothesis is that it is these forces on the iris that caused the pupillary capture. We have clarified this point in the discussion.

NEED TO DO

I am not certain that moving the position of the lens anteriorly by 1 mm (2mm posterior) would have necessarily made it less likely to engage iris.

Response: If the lens is moved anteriorly to the point that is it sitting just posterior to the iris, it is unlikely that the pupillary margin will be captured by the IOL due to the IOL stabilizing the iris and preventing movement. In this way, moving the lens forward may do the same thing.

In the results section of the case description the multiple episodes of OHT is a bit unclear. Understanding the IOP range between episodes as well as the OHT range during capture should be spelled out more clearly. The lower end of the range ("9") is NOT an example of ocular hypertension. Units mmHg.

Response: Thanks for your comment. The IOP ranges have the clarified further.

"As with any surgery, the risks and benefits of sutureless surgery must be considered" -- what are the choices for patients -- suture, suture-less, ACIOL and aphakic CL. List these more clearly.

 Response: These have been added.

Minor comments:

First sentence is not a complete sentence in the abstract (!)

Response: This has been updated

I agree with the authors that exam for floppy iris in important before Yamane undertaken, but how exactly is this done? What are the alt choice for surgery that should be advocated if a floppy iris is identified? Is this complication common enough to justify a different IOL placement technique and if so at what threshold? 

Response: This is done by doing a slit lamp examine and examining the iris with rapid movement of the eye.

Response: We are not saying that an alternative choice of surgery must be undertaken, but rather that noting preoperative floppy iris may help with patient consulting and those patients may benefit from more anterior IOL placement as well as a surgical PI during the time of surgery.

Figure 3 needs revision as gazes are too similar to illustrate the point. If quantifiable (degrees), all the better to add

This has been updated. The degrees are not quantifiable

Reviewer 2 Report

The authors present an interesting case of positional pupillary capture. The manuscript needs extensive editing of English language and style. Some sentences don´t make sense e.g. "Examination also but significant iris movement when patient looked up and down without IOL movement." Some more information about the patient should be given, e.g. did the patient have pseudoexfoliation syndrome, when was the initial surgery, what were the BCVAs, what kind of treatment did the patient get for recurrent IOP spikes? 

Author Response

The authors present an interesting case of positional pupillary capture. The manuscript needs extensive editing of English language and style. Some sentences don´t make sense e.g. "Examination also but significant iris movement when patient looked up and down without IOL movement." Some more information about the patient should be given, e.g. did the patient have pseudoexfoliation syndrome, when was the initial surgery, what were the BCVAs, what kind of treatment did the patient get for recurrent IOP spikes? 

Response: The manuscript has been proof read.  More pre-operative details have been added for the patient. The BCVA during the post-operative report is included. The treatment for recurrent IOP spikes were included.

Round 2

Reviewer 1 Report

Thank you for completing this revision

Minor comments. Please revise / clarify "AA4203 lens" this isolated part number is not sufficient in detail.

Consider revision: "a force upon the iris [causing] the positional pupillary capture."

Consider removing from abstract: "... in similar cases."

Author Response

All updates have been made